# Hypothalamic CRH neurons represent physiological memory of positive and negative experience

Tamás Füzesi [1,2], Neilen P. Rasiah[1], David G. Rosenegger[1], Mijail Rojas-Carvajal [1], Taylor Chomiak [2], Núria Daviu[1], Leonardo A. Molina[2], Kathryn Simone[1], Toni-Lee Sterley[1], Wilten Nicola[1] & Jaideep S. Bains [1,3] ✉

Recalling a salient experience provokes specific behaviors and changes in the physiology or internal state. Relatively little is known about how physiological memories are encoded. We examined the neural substrates of physiological memory by probing CRH[PVN] neurons of mice, which control the endocrine response to stress. Here we show these cells exhibit contextual memory following exposure to a stimulus with negative or positive valence. Specifically, a negative stimulus invokes a two-factor learning rule that favors an increase in the activity of weak cells during recall. In contrast, the contextual memory of positive valence relies on a one-factor rule to decrease activity of CRH[PVN] neurons. Finally, the aversive memory in CRH[PVN] neurons outlasts the behavioral response. These observations provide information about how specific physiological memories of aversive and appetitive experience are represented and demonstrate that behavioral readouts may not accurately reflect physiological changes invoked by the memory of salient experiences.

Behaviors generated in response to specific stimuli are often matched by changes in the physiology or internal state of the organism. Behavioral adjustments and shifts in internal state also occur during the recall of prior salient experience[1–3]. There has been considerable work on identifying key brain cell populations, circuits, and mechanisms that give rise to explicit, behaviorally detectable forms of memory, but surprisingly, little is known about how the brain remembers prior experience to generate the requisite physiological changes.

A widely held but largely untested assumption is that external and internal responses rely on the same neural architecture for memory. In part, this is due to limitations imposed by the slow kinetics of hormonal release and difficulties in accurately sampling peripheral signals in parallel with behavioral observations. To explore the substrates of a physiological memory, we examined corticotropin-releasing hormone-synthesizing neurons in the paraventricular nucleus of the hypothalamus (CRH[PVN]). These cells control the hypothalamic–pituitary–adrenal (HPA) axis[4] and, by

extension, the circulating levels of corticosterone (CORT)[4–6], the classical stress hormone. CRH[PVN] neurons show bi-directional changes in activity to aversive (increase activity) and rewarding (decrease activity) stimuli[7,8], detect predatory threats to gate specific defensive behaviors[9,10], and are required for the generation of anxiety states[11,12]. Repeated optogenetic manipulation of CRH[PVN] activity is sufficient to form contextual memory[7], positioning these cells as potential nodes for storing information about salient experiences.

In rodents, associative memory of threat requires the recruitment of engrams, specific subsets of neurons that are reliably activated during learning and retrieval, described in the cortex, amygdala, and hippocampus[13–16]. However, species with nervous systems containing only rudimentary counterparts or no equivalent structures to these brain regions are still capable of employing associated memory cues[17]. Indeed, oxytocin neurons in the hypothalamus show all the necessary features of a context-dependent fear memory engram[18].

[1]Hotchkiss Brain Institute & Department of Physiology & Pharmacology, University of Calgary, Calgary, Canada. [2]CSM Optogenetics Core Facility, Cumming School of Medicine, University of Calgary, Calgary, Canada. [3]Krembil Research Institute, University Health Network, Toronto, Canada. ✉e-mail: jsbains@ucalgary.ca

Since CRH[PVN] neurons are both necessary[5] and sufficient[11] for the endocrine response to stress, we used in vivo population and single-cell calcium imaging of CRH[PVN] activity to provide detailed, real-time tracking of neurons that control endocrine state over multiple days. We show that CRH[PVN] neurons exhibit contextual memory of both negative and positive stimuli. The negative cellular memories persist for days, indicating that CRH[PVN] neurons track contextual valence in an enduring fashion. Finally, we provide distinct learning rules for positive and negative memories in CRH[PVN] neurons.

## Results

### Sustained anticipatory activity of CRH[PVN] neurons

In order to explore whether CRH[PVN] neurons are affected by the recall of an event with high salience, we first evaluated their activity in response to neutral stimuli. We expressed GCaMP6 in CRH[PVN] neurons and used single fiber photometry and miniature endoscopes (miniscopes) (Fig. 1a–c; Supplementary Fig. 1). Mice were exposed to a novel

context that contained no overt threat cues or appetitive stimuli (neutral valence). CRH[PVN] population activity increased upon exposure to the neutral context, and this increase persisted for the duration of the exposure (Fig. 1d, e). Miniscope recordings also confirmed the activity increase at the level of individual CRH[PVN] neurons during neutral context exposure (Fig. 1f, g). A Principal Component Analysis (PCA) showed both an expansion and a shift in the activity state space in comparison to home cage (Fig. 1h). To determine whether the persistence of the activity increase was due to sustained context exposure and not a prolonged response to the handling during the transfer, mice were handled similarly, but placed back in the home cage. This resulted in an increase in activity during handling, but the activity returned to baseline levels following the return to the home cage (Supplementary Fig. 2). These observations indicate that exposure to context with neutral valence triggers a sustained activity in CRH[PVN] neurons. We labeled this activity anticipatory, since it occurs even in the absence of an overt threat.

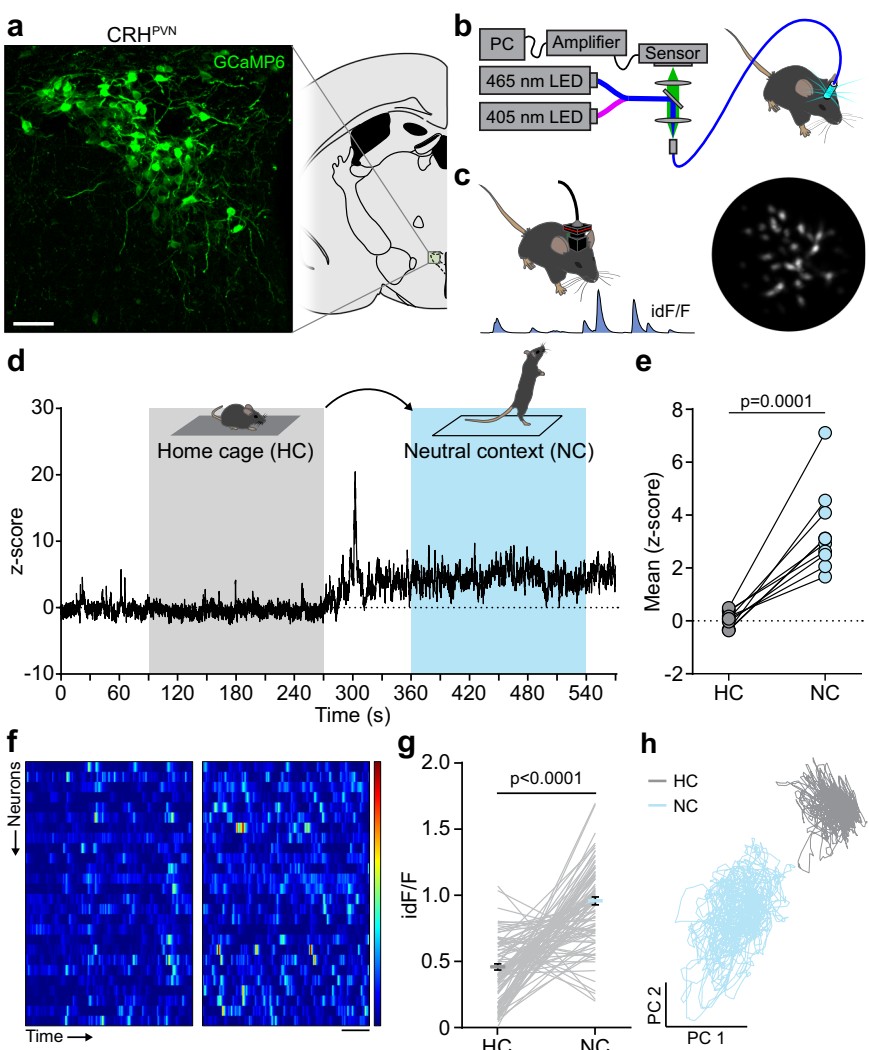

**Fig. 1 | Exposure to a novel, neutral context triggers a sustained response of CRH[PVN] neurons. a** Schematic map and confocal image show the expression of GCaMP6 in CRH[PVN] neurons. Representative example of all neuronal activity recording experiments. **b** Schematic depiction of single fiber photometry recording of CRH[PVN] neurons **c** Left, mouse equipped with a miniature microscope and sample trace from an individual unit to calculate integrated d$F$/$F$ (id$F$/$F$). Right, Miniscope field of view using Min1pipe analysis. **d** Illustrative fiber photometry trace shows the 3 min-long time segments of a home cage (HC) and neutral context

(NC) exposures used for further analysis. **e** Quantification of the mean amplitudes during the different states ($p = 0.0001$, $n = 10$ mice, $t = 6.340$, paired two-tailed $t$-test). **f** Recorded with miniature microscopes, heat maps show the activity of identified neurons during HC and NC exposure. **g** NC exposure on neural activity ($p = 1.057 \times 10^{-17}$, $n = 96$ cells, $N = 4$ mice, $t = 10.56$, paired two-tailed $t$-test). **h** Principal component analysis result shows the relationship of neuronal activities. Scale bars, **a** 20 μm; **f** 30 s. Data are mean ± s.e.m. For further statistical details, see Supplementary Table 1. Source data are provided as a Source Data file.

## Aversive memory encoded by CRH^PVN neurons

To determine whether CRH$^{PVN}$ neurons make a contextual association with intense stress, we exposed mice to ten foot shocks (FS) in a novel environment (Pre) and returned them to this context 24 h later (Post). During the test day, CRH$^{PVN}$ neurons exhibited a more robust increase in activity in comparison to the increase observed prior to FS (photometry: Fig. 2a, b; miniscopes: Fig. 2c, d; Supplementary Fig. 3a). This activity increase was not apparent immediately after foot shock (Supplementary Fig. 3b, c). The increased neural response was accompanied by an increase in CORT that was greater than that observed after re-exposure to the neutral context (Supplementary Fig. 3d). In contrast, re-exposure of mice to a neutral context that was not associated with foot shock triggered the same level of CRH$^{PVN}$ activity, both at the population level (Fig. 2e, f) and at individual neurons (Fig. 2g, h). These data demonstrate that CRH$^{PVN}$ neurons show contextual recall of negative valence.

In order to address how contextual memory is encoded in the hypothalamus, we compared the activity of individual CRH$^{PVN}$ neurons in the context prior to FS (Pre), and in the context the day after FS (Post). There was no correlation between the activity of individual cells in the Pre and Post groups (Fig. 2i). This

suggests that a subpopulation of cells, but not the entire population, contributed disproportionately to the increase in activity. To test whether the subpopulation was assigned randomly or was rule-driven, we clustered CRH$^{PVN}$ neurons using a non-arbitrary data-driven approach to identify an underlying subpopulation structure (Fig. 2j). Three groups were identified based on activity prior to FS; cells that showed little increase in activity were classified as Weak. Cells with the highest activity prior to FS were classified as Strong. The remainder of the population was intermediate. Weak cells showed a dramatic increase in activity in the context on the day after FS; in contrast, the response of Strong cells was similar on both days (Fig. 2k). In fact, during recall, the activities of cells classified as Weak and Strong based on activity prior to FS, were indistinguishable (Fig. 2k). On the other hand, re-exposure to neutral context revealed a high correlation between Pre and Post (Fig. 2l). Furthermore, Weak and Strong clusters identified based on the initial neutral context exposure remained significantly different on neutral context re-exposure (Fig. 2m, n). Additional differences were observed between the Weak and Strong cells in the foot shock experiment: Weak cells had a stronger response to the shocks (Fig. 2o). Importantly, the

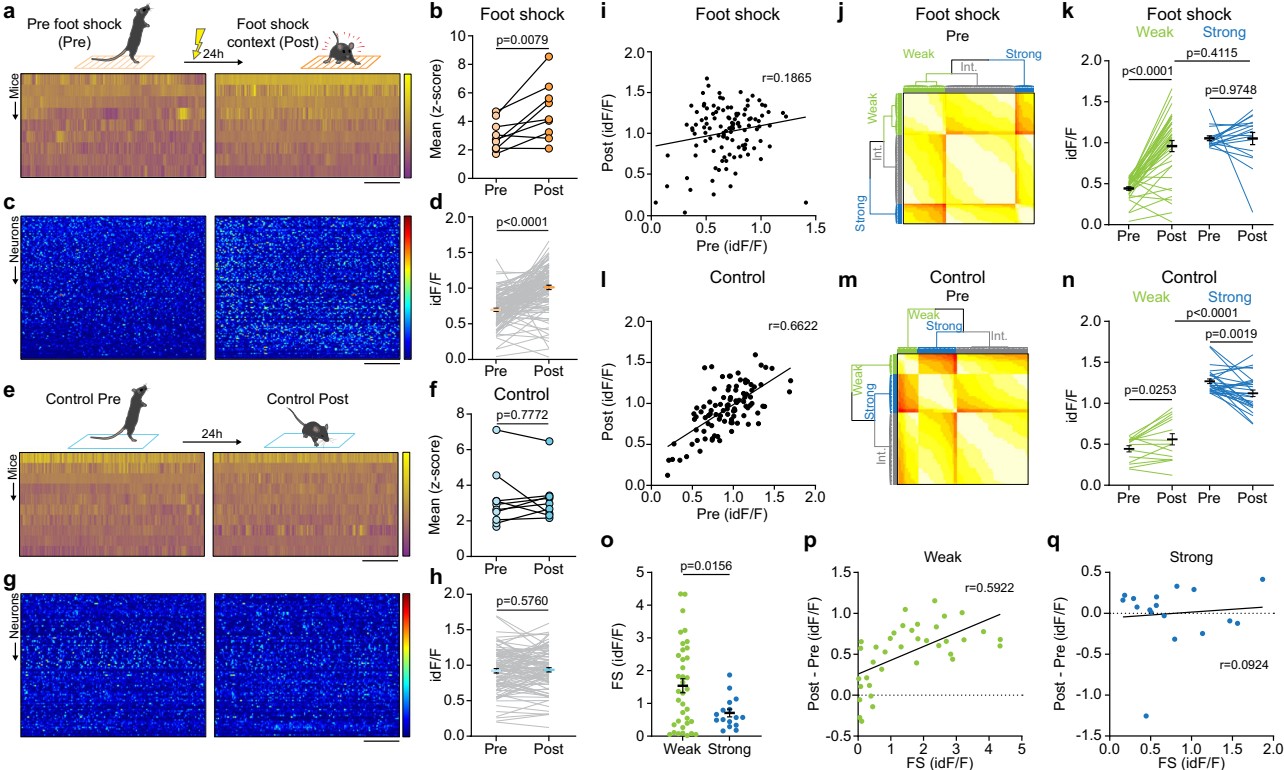

**Fig. 2 | Strong stress exposure updates the activity of the CRH$^{PVN}$ population.**
**a** Top, Experiment schematics. Bottom, Heat maps of fiber photometry recordings. **b** CRH$^{PVN}$ population activity during anticipation (Pre) and context re-exposure following foot shock (Post) ($p = 0.0079$, $n = 9$ mice, $t = 3.514$, paired two-tailed $t$-test). **c** Heat maps of neuronal activity. **d** Neuronal activity quantification ($p = 6.634 \times 10^{-16}$, $n = 115$ cells, $N = 5$ mice, $t = 9.409$, paired two-tailed $t$-test). **e** Top, Control experiment schematics. Bottom, Heat maps of fiber photometry recordings. **f** Population activity of CRH$^{PVN}$ neurons during anticipation (Pre) and neutral context re-exposure (Post) ($p = 0.7772$, $n = 9$ mice, $t = 0.2927$, paired two-tailed $t$-test). **g** Heat maps of neuronal activity. **h** Neuronal activity quantification ($p = 0.5760$, $n = 96$ cells, $N = 4$ mice, $t = 0.5612$, paired two-tailed $t$-test). **i** Correlation of neuronal activity between Pre and Post foot shock ($p = 0.0458$, $r = 0.1865$, linear regression). **j** Data-driven clustering of CRH$^{PVN}$ based on Pre activity. **k** Neuronal activity of Weak and Strong anticipatory neurons (Weak, $p = 2.952 \times 10^{-10}$, $n = 37$ cells, $t = 8.602$, paired two-tailed $t$-test); (Strong, $p = 0.9748$, $n = 17$ cells, $t = 0.0320$,

paired two-tailed $t$-test). Comparison whether the subpopulations are still distinguishable on Post ($p = 0.4115$, $t = 0.8279$, two-tailed $t$-test). **l** Correlation of neuronal activity between Control Pre and Post ($p = 7.5 \times 10^{-14}$, $r = 0.6622$, linear regression). **m** Data-driven clustering of CRH$^{PVN}$ based on Control Pre activity. **n** Neuronal activity Weak and Strong anticipatory neurons (Weak, $p = 0.0253$, $n = 15$ cells, $t = 2.504$, paired two-tailed $t$-test); (Strong, $p = 0.0019$, $n = 28$ cells, $t = 3.450$, paired two-tailed $t$-test). Comparison whether the subpopulations are still distinguishable on Post ($p = 2.670 \times 10^{-05}$, $t = 7.617$, two-tailed $t$-test). **o** Foot shock (FS) response of CRH$^{PVN}$ subpopulations ($p = 0.0156$, $t = 2.501$, two-tailed $t$-test). **p** Correlation between FS and the subsequent activity increase on Post of Weak neurons ($p = 0.0001$, $r = 0.5922$, linear regression). **q** Correlation between FS and the subsequent activity increase on Post of Strong neurons ($p = 0.7244$, $r = 0.0924$, linear regression). Scale bars, **a**, **e** 30 s, −2.5 to 15 z-score **c**, **g** 30 s, 0–15 idF/F. Data are mean ± s.e.m. For further statistical details, see Supplementary Table 1. Source data are provided as a Source Data file.

level of FS response was a strong predictor of the activity increase during context retrieval in Weak cells (Fig. 2p). On the other hand, Strong cells showed no change in activity that could be explained by their FS response (Fig. 2q). These findings were corroborated if we identified Weak and Strong cells based on activity profiles that are more than one standard deviation removed from the mean (Supplementary Fig. 4). These analyses reveal that FS predicts the activity increase best for cells that were the weakest responders prior to FS (Supplementary Fig. 5).

### Learning rule in CRH^PVN neurons

These results imply that two factors are critical to induce a context-dependent change in the threat anticipation of CRH^PVN neurons: the relative weakness of cell activity during context exposure prior to FS and the relative strength of the response to the FS itself. We considered if these two factors alone could explain the update in the CRH^PVN response by simulating networks of plastic, leaky-integrate-and-fire neurons which display a synthetic calcium-like indicator (Fig. 3a; Table 1). These neurons received inputs that represented context weighted by a vector $w_i$ for each simulated CRH^PVN neuron. The weight $w_i$ was plastic, with its changes dictated by a simple learning rule (Fig. 3b) where the weights are updated according to two terms:

$$\Delta w_{ij} \propto \left(\Omega - \sum_i w_{ij}\right) \cdot H\left(\mathrm{Ca}(t) - \mathrm{Ca}^{\{\mathrm{thresh}\}}\right)\mathrm{Ca}(t) \quad (1)$$

The first term models the recruitment of cells. Neurons with lower sums of weights are recruited more strongly to reach the weight ceiling $\Omega$. The second term acts as a plasticity gate, which is opened by strong foot shock responses over a plasticity threshold (for additional details, see methods). We found that these two terms could reproduce the increase in the anticipatory response of CRH^PVN neurons (Fig. 3c, d). This included the ceiling level recruitment from the Strong anticipatory pool (Fig. 3e) and the population level response in the low-dimensional PCA space (Fig. 3f). Collectively, these results

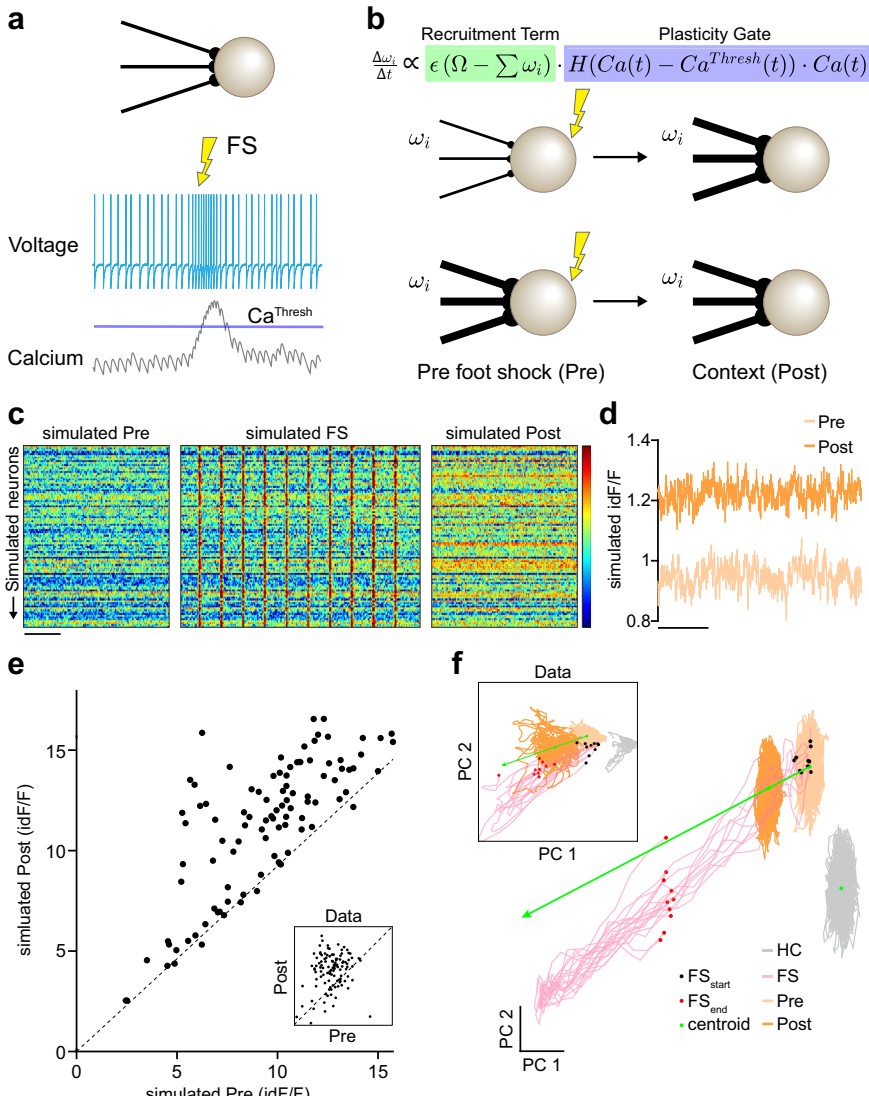

**Fig. 3 | Two-factor authentication rule for endocrine fear memory. a** A network of leaky-integrate and fire (LIF) neurons with random inputs acts for the context effect prior (Pre) and during foot shock (FS) and serves as a model. Each neuron has a calcium-like indicator variable, which can trigger plasticity if it exceeds a threshold. **b** The change in a LIF neuron's input weights is dictated by a two-factor rule. The first factor is modeled on the observed recruitment. Neurons with small weight sums ($\sum w_i$) have lower activities and thus are more strongly recruited by

$\Omega - \sum w_i$. The second factor is the plasticity gate, which is triggered by the foot shock. **c** Simulated calcium indicator in the Pre, FS, and Post. **d** The simulated calcium indicator increases on Post. **e** Change in simulated calcium indicator variable (with data inset). The dashed line shows $x = y$. **f** Principal component analysis plots of the simulated network (with data inset). Scale bar, **c** 60 s, 0–2.4 z-score; **d** 60 s.

**Table 1 | Table of parameters and their values for the spiking neural network simulations**

| Parameter | Value |
|---|---|
| $\tau_m$ | 10 ms |
| $v_{reset}$ | $-65\,mV$ |
| $v_{threshold}$ | $-40\,mV$ |
| $I_{bias}$ | $-41\,pA$ |
| $d$ | $20\,pA$ |
| $\tau_w$ | $100\,ms$ |
| $N$ | $100\,Neurons$ |
| $N^{context}$ | $1000\,Context\,inputs$ |
| $\tau_d^{context}$ | 20 ms |
| $\tau_r^{context}$ | 2 ms |
| $\epsilon$ | $0.014\,s^{-1}\,(FS), 2\times10^{-6}\,s^{-1}\,(Nutella)$ |
| $\Omega$ | $0.0031\,(FS), -1\,(Nutella)$ |
| $\tau_d$ | 1 s |
| $\tau_r$ | 100 ms |
| $\tau_\omega$ | 400 s |

demonstrate that the activity of CRH^PVN neurons is updated following intense stress using a local memory mechanism.

**Uncoupled internal and external representations of stress**

Next, we examined the relationship between this CRH^PVN response and behavior. When returned to the context one day after FS, mice exhibited canonical fear behavior (freezing) (Fig. 4a, b). Consistent with previous reports, repeated exposure to the FS context resulted in the extinction of freezing behavior[13–15] (Fig. 4a, b). In contrast, the CORT response was similar following each context exposure after FS (Fig. 4c, d). Consistent with the CORT response, the CRH^PVN population response to FS context was invariant for the repeated exposures (Fig. 4e) and showed no sign of extinction. Next, we examined the activity profiles of individual cells over multiple days in the FS context. We observed that once individual CRH^PVN cells were recruited to the strong pool, they remained in this pool for multiple days (Fig. 4f). Additionally, the activity of a given cell on one day of FS context re-exposure was a strong predictor of its activity on subsequent days (Fig. 4g, h; Supplementary Fig. 6). PCA further confirmed that the shift in state space occupied by CRH^PVN activity is preserved upon repeated recalls (Fig. 4i). These data demonstrate CRH^PVN neurons track negative valence for many days after an aversive experience. The contextual memory of the aversive experience increases the number of Strong cells by recruiting cells that were Weak.

**Appetitive memory encoded by CRH^PVN neurons**

CRH^PVN neurons show decreases in activity in response to appetitive stimuli[7,8]. Here we asked whether CRH^PVN neurons show evidence for recall of an appetitive stimulus. Following a 5-minute exposure to a novel context, (Pre) a small sample of hazelnut spread was placed in one corner. Consistent with previous results for other appetitive stimuli, access to the hazelnut spread decreased CRH^PVN activity (Supplementary Fig. 7). When mice were returned to the context 24 h later (Post), CRH^PVN activity was significantly lower than recorded in the context before exposure to the hazelnut spread (Fig. 5a–d). In contrast with the effect of foot shock, re-exposure to the hazelnut spread-associated context triggered a CRH^PVN response that was highly correlated with the activity prior to the spread presentation (Fig. 5e). However, data-driven clustering of CRH^PVN neurons showed that Weak cells responded similarly upon contextual recall, while Strong cells showed a dramatic drop in activity (Fig. 5f, g). This blunted CRH^PVN response was evident on 2 additional days of context exposure (Fig. 5h, i). Furthermore, the activity of individual CRH^PVN neurons was similar

and predictable throughout the repeated recalls (Fig. 5i, j). The stability of this response was confirmed with PCA (Fig. 5k). Data simulation confirmed that a simpler, 1-factor learning rule that only depends on Pre activity adequately describes the attenuated CRH^PVN response (see "Methods"; Supplementary Fig. 8).

Our data demonstrate that CRH^PVN cells show scaled responses to stimuli of negative, neutral, and positive valence. Furthermore, they faithfully recall positive and negative valence upon re-exposure to context over multiple days (Fig. 5h).

## Discussion

Our findings demonstrate that hypothalamic CRH^PVN neurons show associative learning to form enduring contextual memories of positive and negative experiences. This learning is local and independent of learning in other brain regions that guide behavioral responses. Negative experience invokes a two-factor learning rule in which additional cells are recruited into a contextual response pool based on their activity prior to the aversive stimulus and the magnitude of their response to the stimulus itself. The spectrum of CRH^PVN neuronal responses to context exposures highlights the diversity of these neurons. This is consistent with recent observations that this population is heterogeneous[19–22]. Unlike other forms of associative learning in which the most active cells are selectively recruited into a memory trace, here, it is the least active cells that are preferentially targeted. Positive or appetitive experience invokes a simpler, 1-factor learning rule in which there is a scaled decrease in the activity of the CRH^PVN neurons upon exposure to the appetitive context.

The anticipation of threat activates a defensive organismic state to recruit specific circuits that elicit appropriate defensive behaviors[23–29]. Threats or dangers, however, are not always predictable. This unpredictability means specific brain systems must also anticipate and prepare for potential danger[1–3]. We propose that the anticipatory activity of CRH^PVN neurons in response to a decrease in safety is a cellular-level, precautionary activity signature. This is consistent with a theoretical security motivation that prepares the organism for the immediate utilization of available resources without necessarily consuming them[1]. We theorize that precautionary activity in CRH^PVN functions as an early permissive signal, informing key decision-making centers about the organismal state.

The fact that changes in the activity of CRH^PVN neurons outlast contextual behaviors suggests this hypothalamic memory center likely functions independently of the cortex, amygdala, and hippocampus, all brain regions that are reliably activated during learning and retrieval[13–16]. The concept of a hypothalamic memory center is consistent with previous work showing the existence of fear memory in hypothalamic oxytocin neurons[18]. Even more broadly, species with a phylogenetically older nervous system are still capable of forming associative memories[17].

For negative valence, we propose a two-factor learning rule that adds a new dimension to classical Hebbian learning rules that link immediate activity increases to a stimulus to learning[16,30]. It is also distinct from previous observations that take neuronal activity into account; for example, in the amygdala, cells with higher activity before training are preferentially recruited into a memory trace following an aversive experience[31]. For positive valence, the rule is simpler, with a scaling down of activity across the entire population during re-exposure to context that is not dependent on changes in neural activity during the presentation of the inducing stimulus.

Finally, our observations reveal a disconnect between behavioral responses and CRH^PVN neurons (and CORT response) during repeated exposure to negative contexts. While the behavioral response shows extinction with repeated context exposure, the activity of CRH^PVN neurons, which we interpret as reflective of an internal state process, remains fixed. Since these internal states are only directly accessible to

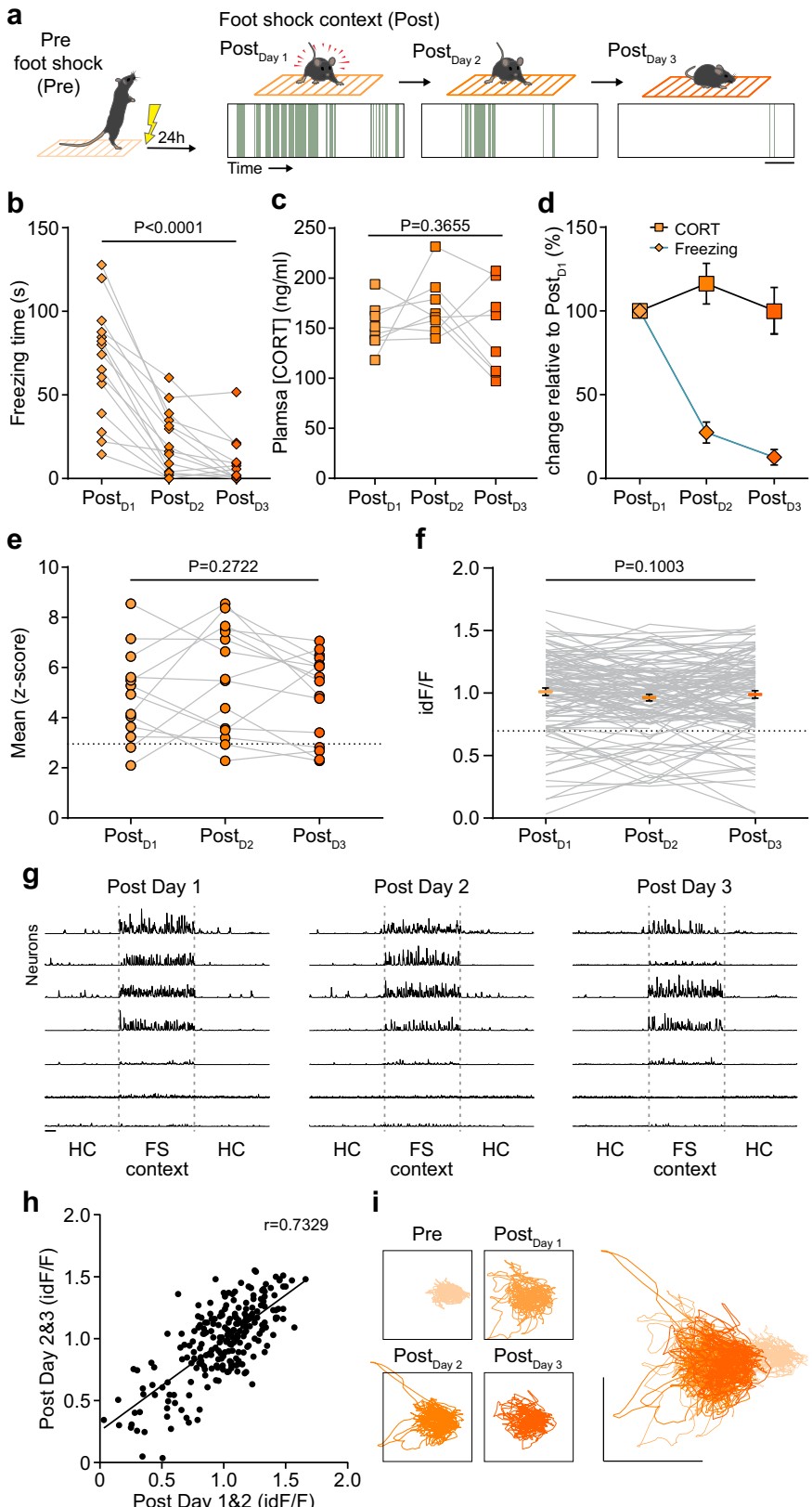

the individual, others must make behavioral-based inferences in order to deduce an individual's emotions[32]. Our data showing a disconnect between a physiological memory and a behavioral memory suggests that the emotional state of the animal may not necessarily be reported faithfully by behavioral readouts. Authoritative voices like Charles Darwin[33,34] and William James[35] have posited that behaviors can be used

to deduce these introspective states. Consistent with these ideas, using specific behaviors as proxies for emotional states has advanced our understanding of the neural basis of anxiety and fear[28]. However, our results showing that CRH[PVN] activity and behavior do not always match suggest that externally visible behaviors may not always reliably report internal states such as stress.

**Fig. 4 | Behavioral and endocrine fear responses diverge. a** Analysis of freezing in the 3-min observation periods during re-exposures to the foot shock context (Post). Illustrative barcodes represent freezing bouts from one animal. **b** Quantification of freezing time ($n = 15$ mice, $p = 5.638 \times 10^{-10}$, $F(2,28) = 50.09$, repeated measures 1-way ANOVA). **c** Corticosterone response to re-exposure ($n = 8$ mice, $p = 0.3655$, $F(2,14) = 1.082$, repeated measures 1-way ANOVA). **d** Summary of the relative change of behavior and CORT during repeated exposure derived from panels (**b**) and (**c**). **e** Mean of population activity recorded with fiber photometry during each exposure epoch to foot shock (FS) context re-exposure. (Post$_{Day\,1}$–Post$_{Day\,3}$: $P = 0.2722$, $F(2,26) = 1.369$, mixed effects model ANOVA). The dashed line shows the mean activity during Pre. **f** Activity of identified units is similar during all repeated exposures to the FS context (Post$_{Day\,1}$–Post$_{Day\,3}$: $p = 0.1003$, $n = 115$ cells, $N = 5$ mice, $F(2,228) = 2.323$, repeated measures 1-way ANOVA). The dashed line shows the mean activity during Pre. **g** Sample activity traces of identified neurons over multiple days of FS context re-exposure. **h** Correlation of neuronal activity between subsequent days of context exposure ($p < 10^{-15}$, $r = 0.7329$, linear regression). **i** Principal component analysis result shows the relationship of neuronal activities between Pre, Post$_{Day\,1}$, Post$_{Day\,2}$, and Post$_{Day\,3}$, overlayed on the right. Scale bar, **a**, **g** 30 s, Data are mean ± s.e.m. For further statistical details, see Supplementary Table 1. Source data are provided as a Source Data file.

## Methods

### Mice
All animal protocols were approved by the University of Calgary Animal Care and Use Committee (AC21-0067). For fiber photometry and behavioral tracking experiments *Crh-IRES-Cre* (*B6(Cg)-Crhtm1(cre)Zjh/J*; stock number 012704) and *Ai14* (*B6.Cg-Gt(ROSA)26Sortm14(CAG-TdTomato)Hze/J*; stock number 007914) mice in which CRH neurons express tdTomato fluorophore were used[36]. For some fiber photometry and all of the miniature microscopy experiments, the offsprings of *Crh-IRES-Cre* mice crossed with *Ai148 (Ai148(TIT2L-GC6f-ICL-tTA2)-D*; stock number 030328) animals were utilized. Mice were obtained from Jackson Laboratories.

Mice were housed on a 12-h:12-h light:dark cycle (lights on at 7:00 a.m.) with ad libitum access to food and water in whole litters until 1–2 d before use, then were individually housed during the experimental phase. All subjects were randomly assigned to different experimental conditions used in this study. Mice were 6–8 weeks old at the time of surgery and virus injection.

### Stereotaxic injection and optical fiber/GRIN lens implantation
Toally, 6–8 weeks old *Crh-IRES-Cre;Ai14* mice were maintained under isoflurane anesthesia in the stereotaxic apparatus. For fiber photometry experiments, a glass capillary containing a Cre-dependent AAV construct with GCaMP6s (AAV9-CAG.Flex.GCaMP6s; Penn Vector Core) was lowered into the brain (anteroposterior (AP), −0.7 mm; lateral (L), −0.3 mm from the bregma; dorsoventral (DV), −4.5 mm from the dura). The virus was pressure injected with a Nanoject II apparatus (Drummond Scientific Company) in a total volume of 210 nl. Two weeks were allowed for recovery; subsequently, a 400 μm diameter mono fiber optic cannula (Doric Lenses, MFC_400/430/0.48_5mm_MF2.5_FLT) was implanted dorsal to the PVN. For the miniature microscopy experiments, the GRIN lens (6.1 mm length; Inscopix) was lowered dorsal to the PVN of *Crh-IRES-Cre;Ai148* mice at a 100 μm/min speed using a motorized stereotaxic apparatus. Both implantations were targeted to a similar position (AP, −0.7 mm; L, −0.2 mm from the bregma; DV, −4.2 mm from the dura) and were affixed to the skull with METABOND® and dental cement. For photometry, mice were given 2 weeks to recover prior to the start of the experiment. For miniature microscopy, at least one month after lens implantation, a baseplate was installed on the head. For experiments with the hazelnut spread, lenses with integrated baseplates were implanted. Experiments started after an additional two weeks of recovery and handling.

### Histology
To verify GCaMP expression and ferrule location, following the experiments, mice were anesthetized with sodium pentobarbital (30 mg/kg) and transcardially perfused with phosphate-buffered saline (PBS), followed by 4% paraformaldehyde (PFA) in phosphate buffer (PB, 4 °C). Brains were placed in PFA 24 h followed by 20% sucrose PB. In total, 30 μM coronal brain sections were obtained via cryostat. Slide-mounted and coverslipped sections were imaged using a confocal microscope (Olympus BX50 Fluoview). Images were prepared by ImageJ.

### Fiber photometry recording
Fiber photometry was used to record calcium transients from CRH neurons in the PVN of freely moving mice. After the recovery period, animals were handled for 5 min a day for 3 successive days and then habituated to the optic fiber in their homecage (15 min a day) for 3 additional days. We recorded 10 min of CRH$^{PVN}$ neuron activity in the homecage immediately before and after each test for better bleaching correction and the first and last 5 min of recording were excluded from further analysis. The data obtained from injected GCaMP6s (*Crh-IRES-Cre;Ai14*) vs GCaMP6f (*Crh-IRES-Cre;Ai148*) expressing mice were indistinguishable (Supplementary Fig. 9a).

Two different fiber photometry systems were used. TDT system: Briefly, two excitation LEDs (470 nm M470F3 and 405 nm M405F1 from THORLABS) were controlled by a RZ5P (Tucker-Davis Technology) processor running Synapse software (Tucker-Davis Technology). The LEDs were modulated at 211 Hz (470 nm) and 531 Hz (405 nm) to avoid contamination from room lighting. Both LEDs were connected to a DORIC Mini Cube filter set (FMC4_AE(405)_E(460-490)_F(500-550)_S) and the excitation light was directed to the animal via a mono fiber optic patch cord (Doric MFP_400/460/900-0.48_2m_FC/MF2.5). The power of the LEDs was adjusted to provide 30 μW at the end of the patch cord. The resulting signal was detected with a photoreceiver (Newport model 2151) and demodulated by an RZ5P processor. Doric fiber photometry system: Consisting of two excitation LEDs (465 nm and 405 nm from Doric) controlled by an LED driver and console, running Doric Studio software (Doric Lenses). The LEDs were modulated at 208.616 Hz (465 nm) and 572.205 Hz (405 nm), and the resulting signal was demodulated using lock-in amplification. Both LEDs were connected to a Doric Mini Cube filter set (FMC5_E1(405)_E2(460-490)_F1(500-550)_S) and the excitation light was directed to the animal via a mono fiber optic patch cord (DORIC MFP_400/460/900-0.48_2m_FC/MF2.5). The power of the LEDs was adjusted to be 30 μW at the end of the patch cord. The resulting signal was detected with a photoreceiver (Newport model 2151).

### Fiber photometry data analysis
Fluorescent signal data was acquired at a sampling rate of 1 kHz (TDT system) and 100 Hz (Doric system). Data were then exported to MATLAB (Mathworks) for offline analysis using custom-written scripts. Briefly, the 465/470 nm and 405 nm data were first individually fit with a second-order curve which was then subtracted to remove any artifacts due to bleaching. Next, a least-squares linear fit was applied to the 405 nm to align it with the 470 nm channel and then the change in fluorescence ($\Delta F$) was calculated by subtracting the 405 nm Ca$^{2+}$ independent baseline signal from the 470 nm Ca$^{2+}$ dependent signal at each time point. This approach mitigated the differences arising from the lowering recorded voltage due to the continuous bleaching of the light path (Supplementary Fig. 9b, c). In case the motion artifact was too large and could not be eliminated by the subtraction of the 405 signal, the entire recording was discarded. To minimize the impact of the handing and the upstate on the bleaching correction, the first and the last 5 min of fiber photometry recording were used for curve fitting (Supplementary Fig. 9b). For analysis, the z-score calculation

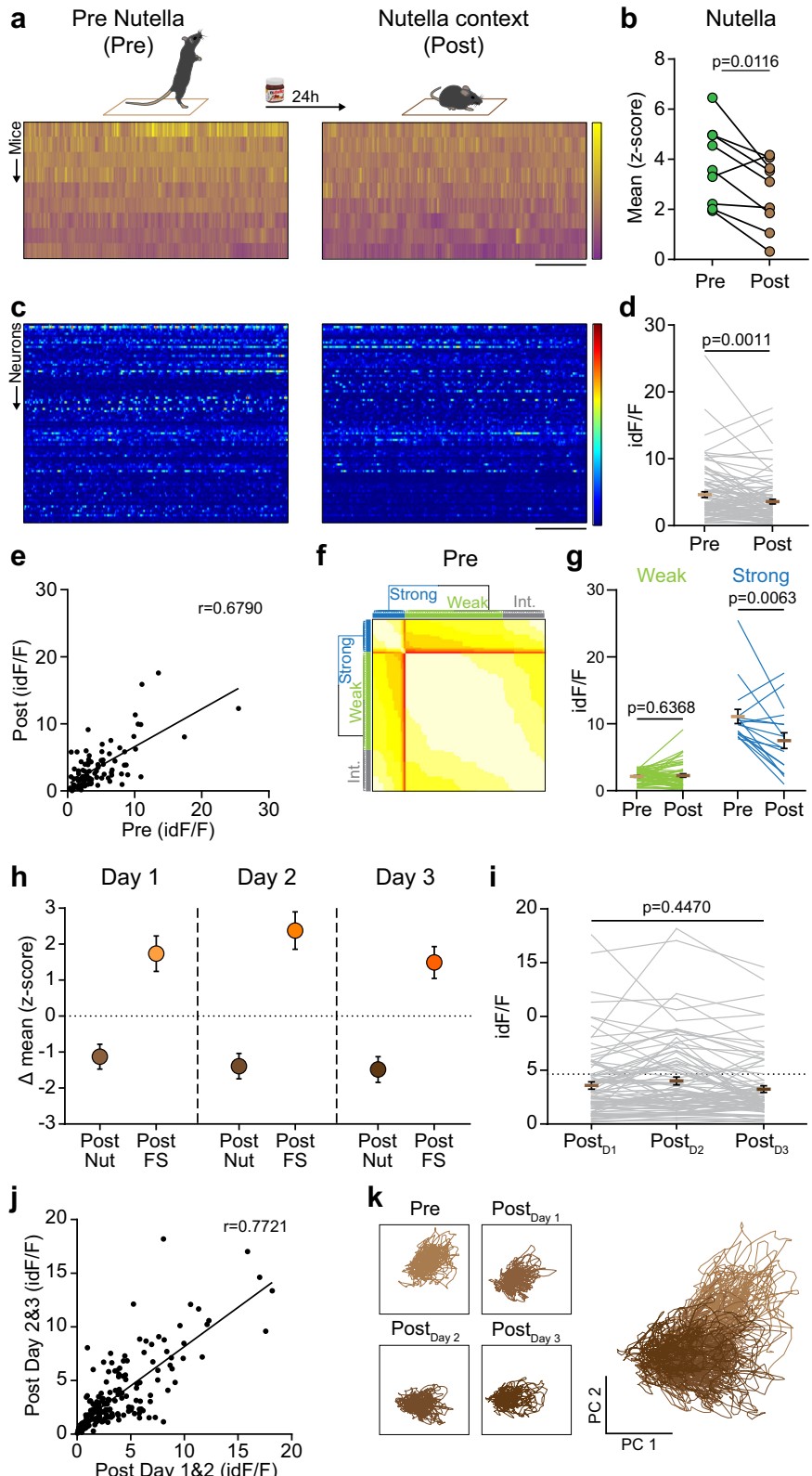

was performed using the following equation $z = (F - F_0)/\sigma F$. Where $F$ is the test signal, $F_0$ and $\sigma F$ are the mean and standard deviation of the baseline signal, respectively. For recordings from the same animals on consecutive days, in order to minimize the distortions arising from different baseline activities on different days, we calculated the $z$-score using the standard deviation from the baseline period of the first day of the experiment (Supplementary Fig. 10a). On each day a 3-min baseline period in the home cage prior placing the animal to a different context was used to calculate relative change triggered by context exposure. To exclude the possibility that slow changes in GCaMP expression or basal CRH$^{PVN}$ activity that affect both baseline and context exposure recording caused the observed changes during repeated exposures, we performed the fiber photometry analyses using methods that are insensitive to the baseline and used relative differences exclusively.

**Fig. 5 | Exposure to appetitive stimulus attenuates the contextual activity of CRH$^{PVN}$ population. a** Top, Schematics of the experiment. Bottom, Heat maps of fiber photometry recordings show the CRH$^{PVN}$ population activity during anticipation before presenting the hazelnut spread (Pre) and context re-exposure (Post). **b** Quantification of the mean amplitudes during the different states ($p = 0.0116$, $n = 9$ mice, $t = 3.257$, paired two-tailed $t$-test). **c** Heat maps show the activity of identified neurons. **d** Activity of identified units during Pre and Post ($p = 0.0011$, $n = 90$ cells, $N = 5$ mice, $t = 3.362$, paired two-tailed $t$-test). **e** Correlation of neuronal activity between Pre and Post ($p = 1.92 \times 10^{-13}$, $r = 0.6790$, linear regression). **f** Data-driven clustering of CRH$^{PVN}$ subpopulations based on Pre activity. **g** Neuronal activity in Pre and Post of Weak and Strong anticipatory neurons (Weak, $p = 0.6368$, $n = 51$ cells, $t = 0.4751$, paired two-tailed $t$-test); (Strong, $p = 0.0063$, $n = 17$ cells, $t = 3.144$, paired two-tailed $t$-test). **h** Relative changes of population activity

recorded using single fiber photometry during all re-exposures to the hazelnut (Nut) context (Post$_{Day 1}$–Post$_{Day 3}$: $p = 0.3302$, $F(2,16) = 0.1188$, repeated measures 1-way ANOVA) and foot shock context. Exposures to foot shock (FS) context in orange are shown as a reference (Nut vs FS: $p = 2.063 \times 10^{-06}$, $F(1,16) = 52.05$, 2-way ANOVA). **i** Activity of identified units during repeated exposures to the hazelnut context (Post$_{Day 1}$ vs Post$_{Day 3}$: $p = 0.4470$, Bonferroni's multiple comparisons test; $p = 0.0055$, $n = 90$ cells, $N = 5$ mice, $F(2,178) = 5.368$, repeated measures 1-way ANOVA). **j** Correlation of neuronal activity between subsequent days of context exposure ($p < 10^{-15}$, $r = 0.7721$, linear regression). **k** Principal component analysis result shows the relationship of neuronal activities between Pre, Post$_{Day 1}$, Post$_{Day 2}$, Post$_{Day 3}$, overlayed on the right. Scale bar, **a** 30 s, −2.5 to 15 $z$-score; **c** 30 s, 0–75 id$F/F$, Data are mean ± s.e.m. For further statistical details, see Supplementary Table 1. Source data are provided as a Source Data file.

Photometry d$F/F$ was calculated by using least-squares second-order polynomial fit. We fitted the 405 nm channel to the 465 nm channel. Then performed the following calculation[37]: d$F/F = (F_{465} − F_{\text{fitted 405}})/F_{\text{fitted 405}}$. Additionally, we performed the formerly described $z$-score analysis using the standard deviation calculated from the home cage baseline activity. All analyses led to the same conclusions (Supplementary Fig. 10b–i).

## Miniature microscopy data analysis

The activity of CRH$^{PVN}$ neurons was recorded continuously for at least 15 min at 20 FPS, 40% LED power, and 2.5 gain with an nVista or nVoke miniature microscope (Inscopix) using the nVista Acquisition Software (Inscopix). The video files were cropped with ImageJ and the data analysis was performed by the MIN1PIPE script[38] in MATLAB using 0.25 spatial and 0.5 temporal downsampling ratio and manual selection of ROIs. idF/F was calculated by averaging the dF/F results sampled at 0.1 s over a 3-min long period and multiplied by 100.

## Calcium imaging alignment via MIN1PIPE

The maximal projection image from MIN1PIPE, along with the assigned and curated putative cells, are used to determine the alignment across all days. A stochastic descent algorithm is applied to both images to maximally align the images across recording days. In short, the images are randomly rotated, translated, and re-scaled to minimize the average pairwise Frobenius distance between all possible pairs:

$$\text{GA} = \sum_{k=1}^{n_{\text{days}}} ||\boldsymbol{X}^k - \boldsymbol{Y}^k|| = \sum_{k=1}^{n_{\text{days}}} \sqrt{\sum_{j=1}^{n} \sum_{i=1}^{m} \left( X_{ij}^k - Y_{ij}^k \right)^2} \quad (2)$$

The random translations, rotations, and re-scaling transformations are applied to pairs of images within all recording days, randomly selected. The Global alignment metric, GA is computed for both the cluster assignments (where cells exist in the imaging plane) and the raw images. A weighted linear combination of the two is then used as an optimization criterion with weights of 0.3 (clusters) and 0.7 (raw). If this weighted sum decreases with an image transform, the image transform is preserved. New image transforms are then randomly created, perturbing off this image in subsequent steps. This stochastic algorithm is performed for 10,000 iterations, with the perturbation size halved every 500 iterations and the pair of recording days within the set swapped every 400 step sizes. Several iterations of this stochastic approach were performed with different initial conditions to ensure both the stability and quality of the cell-to-cell mapping across days (Supplementary Fig. 9d, e).

## Behavioral tests

Following fiber or GRIN lens implantation, mice were individually housed for at least 2 weeks before behavioral testing. Prior to any behavioral manipulation, mice were handled for 5 min each day for 3 successive days and habituated to the experimental room 1 h prior to testing. The apparatus was cleaned with a 70% ethanol solution to eliminate odor from other mice. All behaviors were recorded and scored offline by an experimenter blind to the treatment. Behavioral annotations were performed with a custom-written Excel macro[11]. Freezing behavior was noted if the animal showed no movement for at least 3 s except for respiratory movements.

**Neutral context exposure.** Following the baseline recording in the home cage, mice were placed in a 41 cm × 19 cm × 20.5 cm clean and transparent plastic arena for 5 min. The second exposure was on a consecutive day.

**Different context exposures.** Following the baseline recording in the home cage, mice were placed either in a 45 cm × 45 cm, semi-transparent walled hole-board or a 53 cm × 53 cm opaque open field.

**Footshock stress.** Following 5 minutes of initial exposure to the footshock chamber, a series of foot shocks were delivered to the mouse (0.5 mA, 2 s shocks delivered every 30 s in a period of 5 min; SMSCK, Kinder Scientific). Mice were re-introduced to the footshock chamber for 5 min on 3 subsequent days.

**Hazelnut spread.** In order to familiarize the animals with the object, the mice received a stick in their homecage a day before the experiment. After 5 min of initial exposure to a context, a small amount of Nutella on a stick was placed into the context. Mice were re-introduced to the same chamber for 5 min on 3 subsequent days.

## Corticosterone measurement

Immediately before and 15 min after the onset of stress exposure, blood from the tail vein was collected into ice-cold EDTA capillary tubes (Stardest) and centrifuged (Eppendorf centrifuge 5430 R,7335 g, 4 °C, 20 min). Aliquots of plasma were stored at −20 °C until assayed using a DetectX Corticosterone Immunoassay kit (Arbor Assay). Plasma samples were run in triplicates on the same day.

## Statistics

GraphPad Prism 10.0 software was used for statistical analysis. When comparing means from two dependent groups, different time points paired $t$-test (two-tailed) were used. When comparing the means of two independent groups, unpaired $t$-test (two-tailed) were used. When comparing the means of multiple groups, parametric repeated measures, mixed effects model, and one-way or two-way ANOVA were used, followed by Bonferroni corrections for multiple comparisons. For further statistical details regarding each analysis, see Supplementary Table 1.

To identify subpopulations of neurons based on activity, affinity propagation clustering was used[39,40]. The first step consisted of affinity propagation to reveal data-point exemplars contained in the data using the Euclidean distance metric and an input preference set for the minimum of input similarities ($q = 0$)[39,40]. The second step used exemplar-based agglomerative clustering on the affinity propagation result. This provides an additional tool for finding the most likely

underlying cluster structure as it is geared toward the identification of meaningful data partitions[39].

## Dimension reduction/PCA

To perform the principal component analysis (PCA), the data vector was constructed by taking the time-aligned intervals of either the neutral context (across days), the foot shock condition (across days), and the Nutella condition (across days). The activity of the neurons was tracked across days (see below), with the data matrix corresponding to

$$X_{ij} = r_i(t_j), i = 1, 2, \dots N, j = 1, 2, \dots M \quad (3)$$

Which corresponds to recorded calcium from neuron $i$ at time $t_j$. As the neurons are aligned across days, and aligned in time intervals, the data was pooled across animals. The time indices $t_1, t_2, \dots t_M$ correspond to the Calcium recordings with a sample rate of 0.1 s and cover multiple acquisition days (the home cage/neutral context, home cage/foot shock chamber, and home cage/Nutella application chamber were pooled in time). PCA was then applied to the matrix $\boldsymbol{X}$ directly via the singular value decomposition.

## Spiking neural network simulations

A network of 100 leaky integrate-and-fire neurons with spike frequency adaptation was used as the model for the individual CRH[PVN] neurons (Table 1):

$$\tau_m \frac{dV_i}{dt} = -V_i + I_{\text{bias}} + I_i^{syn}(t) - w_i(t) + \gamma_i(t) \quad (4)$$

$$\frac{dw_i}{dt} = -\frac{w_i}{\tau_w} + d \sum_{t_{ik} < t} \delta(t - t_{ik}) \quad (5)$$

A neuron with index $i$ is said to fire a spike when the voltage reaches a threshold potential ($V_i(t) = V_{th}$), after which the neuron is reset to $V_i(t) = V_{\text{reset}}$. The adaptation variable, $w_i(t)$, for each neuron, $i$, is incremented by the variable $d$ each time that neuron fires a spike with $t_{ik}$ denoting the $k$th spike fired by the $i$th neuron. The membrane time constant, $\tau_m$ controls the ability of a neuron to filter/remember its inputs while the bias current, $I_{\text{bias}}$ sets the baseline firing rate for a neuron. Each neuron receives a white noise current $\gamma_i(t)$ with mean 0 and standard deviation $\sigma$. The dynamically varying current, $I_i^{syn}(t)$ decomposes as follows:

$$\begin{aligned} I_i^{syn}(t) &= I_i^{\text{context}}(t) + I_i^{FS}(t) + I_i^{\text{Nut}}(t) \\ &= \boldsymbol{\omega}_i^T \boldsymbol{r}^{\text{context}}(t) + \omega_i^{FS} FS(t) + \omega_i^{\text{Nut}} \text{nut}(t) \end{aligned} \quad (6)$$

The weights $\boldsymbol{\omega}_i$ are $N^{\text{context}} \times 1$ vectors that multiply a vector of inputs, the context vector $\boldsymbol{r}^{\text{context}}(t)$. This vector is generated as a $N^{\text{context}} \times 1$ filtered spike train. The spike train is generated from a Poisson process with mean $\nu = 8$ Hz when the simulated animal is in a novel environment, and $\nu = 1$ Hz when the simulated animal is in its home cage. The context spike train is filtered with a 20 ms decay time and 2 ms rise time with a double exponential filter.

$$r_i^{\text{context}}(t) = \sum_{t_{ik}^{\text{context}} < t} \exp\left(-\frac{(t - t_{ik}^{\text{context}})}{\tau_D^{\text{context}}}\right) - \exp\left(-\frac{(t - t_{ik}^{\text{context}})}{\tau_R^{\text{context}}}\right) \quad (7)$$

where $t_{ik}^{\text{context}}$ is the $k$th spike fired by the $i$th context input.

The scalar $\omega_i^{FS}$ is a random weight for an input that mimics the impacts of a footshock:

$$FS(t) = \sum_{i=1}^{10} \exp\left(-(t - t_i^{FS})^2\right) \quad (8)$$

where $t_i^{FS}$ is the $i$th foot shock time. The weights $\omega_i^{FS}$ are each generated from a normal distribution with a mean of 2 and a standard deviation of 1. Note that all weights in this work are unitless, as the filtered spike trains have units of $pA$.

The scalar $\omega_i^{\text{Nut}}$ is a random weight for an input that mimics the application of the hazelnut spread.

$$Nut(t) = \exp\left(-\frac{(t - t^{\text{Nut}})^2}{10^4}\right) \quad (9)$$

The parameter $t^{\text{Nut}} = 930\,s$ was used while the weight $\omega_i^{\text{nut}}$ was drawn from a uniform distribution on the interval $[-0.09, 0.01]$, thereby mimicking the observed inhibitory impacts of the hazelnut spread on Calcium.

All spikes are filtered with an auxiliary variable, $r_i^{\text{slow}}(t)$ which is used to implement the two-factor learning rule below. The filter is a double exponential filter:

$$r_i^{\text{slow}}(t) = \sum_{t_{ik} < t} \exp\left(-\frac{(t - t_{ik})}{\tau_D}\right) - \exp\left(-\frac{(t - t_{ik})}{\tau_R}\right) \quad (10)$$

where $\tau_D$ and $\tau_R$ are the rise and decay times of the double exponential filter.

All equations are integrated with a forward Euler method with a 1 ms time step. The total simulation time is 2190 s, with the initial exposure to the novel environment occurring on the interval [500,1290] s, with 10 foot shocks applied on the interval [900,1200], with 30 s pauses between foot shocks. The context is changed back to the home cage at 1290 s, with a second exposure to the novel context at 1790 s.

## Two-factor learning rule

The context weights, $\boldsymbol{\omega}_i$ are updated according to a two-factor, delayed learning rule:

$$\frac{d\boldsymbol{a}_i}{dt} = \epsilon\left(\Omega - \sum_i \boldsymbol{\omega}_i\right) r_i^{\text{slow}}(t) H\left(r_i^{\text{slow}}(t) - r_{\text{thresh}}\right) \quad (11)$$

$$\tau_\omega \frac{d\boldsymbol{\omega}_i}{dt} = -\boldsymbol{\omega}_i + \boldsymbol{a}_i \quad (12)$$

The first term implements the plasticity rule whenever the auxiliary rate-like variable $r_i^{\text{slow}}(t)$ in the $\boldsymbol{a}_i(t)$ variables. These phenomenologically implement a delay in the plasticity, as the molecular biology behind synaptic weight changes take time. A similar, albeit simpler 1-factor delayed learning rule was sufficient to model the activity changes when Nutella was applied:

$$\frac{d\boldsymbol{a}_i}{dt} = \epsilon\left(\Omega - \sum_i \boldsymbol{\omega}_i\right) H(nut(t) - nut_{thresh}) \quad (13)$$

$$\tau_\omega \frac{d\boldsymbol{\omega}_i}{dt} = -\boldsymbol{\omega}_i + \boldsymbol{a}_i \quad (14)$$

## Calcium imaging alignment via MIN1PIPE

The maximal projection image from MIN1PIPE, along with the assigned and curated putative cells are merged into a two-channel image. A stochastic descent algorithm is applied to the merged images to maximally align the images across recording days. In short, the images are randomly rotated, translated, and re-scaled to minimize the distance.

## Reporting summary

Further information on research design is available in the Nature Portfolio Reporting Summary linked to this article.

## Data availability

The raw data that support the findings of this study are available from the corresponding author upon request. The data generated in this study are provided in the Source Data file.

## Code availability

Scripts used to analyze fiber photometry are deposited here: https://github.com/leomol/FPA; https://doi.org/10.5281/zenodo.5708470. The code for simulating the spiking neurons under the foot shock and the hazelnut spread conditions can be found on ModelDB[41], under accession number 2015420 (https://modeldb.science/2015420). Scripts used to align miniscope recordings are available upon request.

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

## Acknowledgements

We thank Mrs. Cheryl Breiteneder, Ms. Mio Tsutsui, Mr. Carlos Martinez, and Mr. Rodney Barasi for their expert technical support. We also thank Dr. Jinghao Lu for their support with MIN1PIPE. We are grateful for the support of the Cumming School of Medicine Optogenetics Core Facility. This work was supported by an operating grant to J.S.B. from the Canadian Institutes for Health Research (FDN-148440) and the Brain Canada Neurophotonics Platform. ND received support from Alberta Innovates-Health Solutions and the Brain and Behavior Research Foundation.

## Author contributions

T.F. designed and conducted the experiments, analyzed the data, and wrote the paper. N.P.R., M.R.C., D.G.R., N.D. and T.S. conducted experiments, analyzed data, and contributed to paper preparation. K.S., L.A.M, and T.C. contributed to data analysis. W.N. performed the computational modeling and contributed to paper preparation. J.S.B. designed experiments, prepared the paper, and supervised the project.

## Competing interests

The authors declare no competing interests.
