## [Peer Review File · Nature Communications]

REVIEWER COMMENTS

Reviewer #1 (Remarks to the Author):

This manuscript demonstrated that the internal state (endocrine responses) showed an uncoupled relationship with the external state (behaviors) regarding neutral and negative contexts. The authors exposed animals to a neutral environment (open field without overt threat) and a negative context (with foot shock supplied) as two different paradigms, by which they proved that the endocrine level could outlast the habituated behaviors over timescales. Also, they demonstrated that CRH neuronal activity can be influenced by two factors in a negative context: activity prior to stimulus and “strength” of the response to stimulus per se. They concluded the anticipatory activity of CRH neurons provided a readout for the uncoupling status, suggesting that the visible behaviors may not reliably reflect the internal state. However, there’re several concerns with regard to the premise, significance, and conclusion.

Major comments:

1. The issue of the relationship between endocrine state and behaviors. In this paper, the authors used the CORT level as a readout of the HPA axis and tried to link it to the behavioral state. They claimed CORT level was not altered during a neutral context (Fig 1c), while it has a greater increase in a negative context (Fig. 3c). Thus, they thought CRH neurons showed a contextual recall of negative stimuli. But the author also mentioned FS induced a larger increase of CRH neuronal activity than neutral context, and behavioral state changes faster than endocrine effect, so CORT secretion kinetics can be masked by lack of frequent sampling. Thus, the conclusion of the CRH neurons playing a role in a negative context but not in a neutral context is lack strong evidence. I don’t think the authors have given rationale and logical inferences based on convincing evidence.
2. The authors used CORT as an internal state to study its relationship or coupling with the behavioral state. Also in many figures, they used CORT as an internal index to interpret results and conclude the CORT level outlast the behaviors. But they didn’t show any the HPA axis-induced CORT secretion and behavioral changes are independent of each other or that they have a causal relationship somehow, which is a very important premise and question to answer in this study. They should prove whether the uncouple of timescales of the internal and external state is a combinational effect from the circuit and endocrine or just one of them.
3. The issue of fiber photometry. In most of the recording results, authors showed a period of calcium transient traces before and after stimuli, such as HC vs. NC (Extended Fig. 2d, e, f). Normally, people should show a baseline level of the calcium signals and indicate which time window is used as an offset and what the duration is. Especially sometimes when you do a long-term recording over days, you unplug and plug the ferrules back on the heads, the baseline might change or GCaMP expression intensity might increase gradually, which will have an obvious impact on our results. The authors should show a consecutive trace of each day or indicate how they define the starting times of the home cage and neutral context. I don’t see any detailed info in context or even in the method.

4. As for the mini-scope imaging, authors should indicate the n numbers of mice instead of the cell numbers, or they should increase the quantity for each group to $n \geq 5 - 7$ mice. Also, they should also have some analysis including different individuals to make the results more solid. (e.g. Fig. 2d, Fig.3b)

5. They showed relatively rough single-cell imaging data to emphasize the CRH neurons' responses to neutral or negative contexts. A principal component analysis shows a shift in the activity state in comparison to HC. I'm curious about if the activity state calculation is based on the overall epoch in HC, neutral context, or footshock chamber. They should make their method more comprehensive and make the legend concise enough. The writing should be improved.

6. A follow-up question to comment 5.

What is the correlation between certain behavioral patterns and CRH neuronal activities? Will the behavioral patterns change among different contexts? And what are the calcium signals corresponding to a certain behavioral feature or location in the test arena? In addition to the GCaMP signal changes during the entire stimuli, it'll be more comprehensive and solid if the authors could do a deeper analysis from multiple dimensions.

7. Based on the scope of this study, the authors demonstrated the behavioral and endocrine responses exhibited distinct timescales for tracking neutral and negative valence. Meanwhile, there's been a surge of studies on CRH neurons' effect on reward processing. In terms of a generic mechanism, is CRH neuron activity involved in associative learning of positive valence? They were unable to provide a generalized working model of CRH neurons, which weakens the significance of CRH in linking external and internal states.

8. In the conclusion part, by proposing a learning rule which might add a new dimension to classical learning theory, I don't see more other novelties or contributions of this paper to this field beyond the above statement. And the amount of evidence and results seems weak to me. Overall, it looks like an unestablished work needs to be improved a lot. I don't think it's eligible for publication at this stage.

Minor comments:

1. Method needs to be supplemented with more details.

2. Authors might need to include all the individual points in all graphs in addition to mean \pm s.e.m. information. (e.g. Fig. 1j, 5d)

3. The same types of graphs have different displays in all figures, try to make the style as one to make it more concise (Fig. 1a, 2c, 4c).

Reviewer #2 (Remarks to the Author):

In the current manuscript the authors investigate how behavioral responses to acute or repeated exposure to neutral or aversive stimuli maps to the activity of CRH neurons in the PVN and consequently HPA axis activation and CORT secretion. They observed that while there is substantial behavioral adaptation over several test sessions, the CRH-PVN neuronal activity and subsequently the CORT secretion do not seem to habituate, suggesting an uncoupling of the two. Further, they identify a subpopulation of CRH-PVN neurons that is characterized by low activity to a neutral environment and subsequently strong response to a high-stress foot shock-associated environment. This neuronal phenotype was again stable over time and independent of the behavioral expression of extinction learning. The paper is interesting and the experiments are conducted with a high technical standard. However, I disagree with the interpretation of the data and I believe additional experiments would be needed to substantiate the conclusions.

a. A main criticism is the focus of the authors on negative stimuli. CRH-PVN neurons and the HPA axis are not only activated by stimuli with negative valence. In contrast, there is ample evidence that also stimuli with positive valence that lead to positive emotions like joy or excitement significantly activate the HPA axis and CORT secretion. Due to this indiscriminative nature of HPA axis activity, and therefore also CRH-PVN neuronal activity, I would argue that both are unreliable indicators of emotion, independent of the detail and time-scale of the activity assessment. A main assumption of the authors is that CRH-PVN activity is reflective of the valence of internal states, but they only show this with different levels of negative valence. I believe it would be important to test how a positive stimulus (e.g., a reward) would affect behavior in a neutral context and how this would relate to CORT secretion and CRH-PVN activity.

b. The authors interpret the absence of CORT habituation to the neutral environment as dissociation from the behavioral habituation. However, the CORT response is likely triggered just as much by the disturbance of the animals from their home cage environment and handling by the investigator. Thus, while the memory to the already explored environment will result in behavioral adaptation, the remaining threat of being disturbed and handled by the experimenter will always result in a similar CORT response, just as the regular cage change would. I am not sure how this can be tested experimentally, but it calls into question the main hypothesis and interpretation of the authors. Would the same CRH-PVN cells that respond to a neutral or aversive context not also respond in parallel to the circadian activity of the HPA axis in a home cage setting?

c. Can the authors track if weak and strong CRH-PVN cells have different inputs and are preferentially innervated by different cell populations / brain regions? Mapping this potentially differential input network could result in additional insights in the contribution of these two neuronal sub-populations in encoding stressful stimuli.

General Response to Reviewers:

We are grateful to both reviewers for their thoughtful evaluations of our manuscript. They raised a number of important points and in considering their concerns, it became apparent that, in addition to requiring additional data, there were a number of issues in the organization of the manuscript. We now submit a revised version that addresses all of the concerns raised with new experiments and a shift in focus away from the dichotomy between behavior and internal state, a decrease in reliance on CORT measurements and a sharper focus on a local form stress memory in the hypothalamus that controls CRH neuron output. Importantly, we now include data on both negative and positive valence stimuli. These new observations have resulted in a sharper focus on memory mechanisms in CRH^{PVN} neurons, and in combined with reviewer concerns that we did not effectively establish the link between physiological/internal state and emotion, have necessitated a change in the title of the paper.

REVIEWER COMMENTS

Reviewer #1 (Remarks to the Author):

Major comments:

1. The issue of the relationship between endocrine state and behaviors. In this paper, the authors used the CORT level as a readout of the HPA axis and tried to link it to the behavioral state. They claimed CORT level was not altered during a neutral context (Fig 1c), while it has a greater increase in a negative context (Fig. 3c). Thus, they thought CRH neurons showed a contextual recall of negative stimuli. But the author also mentioned FS induced a larger increase of CRH neuronal activity than neutral context, and behavioral state changes faster than endocrine effect, so CORT secretion kinetics can be masked by lack of frequent sampling. Thus, the conclusion of the CRH neurons playing a role in a negative context but not in a neutral context is lack strong evidence. I don't think the authors have given rationale and logical inferences based on convincing evidence.

The reviewer raises a number of important issues here, but the comments also indicate that we did not do our best to explain ourselves. We will try to break down the comments and address each one carefully here:

The idea is that foot shocks induce a type of local memory in this system. We suggest it is local because freezing behaviour, which is classically associated with learning in the amygdala, begins to show extinction upon repeated exposure. This effectively dissociates the CORT response and the behavioral response. As both reviewers have noted, and as we indicated in the manuscript, CORT measurements are not always reliable and can suffer from a temporal disconnect with the actual experience. Hence, we used fiber photometry and miniscopes to get real-time readouts from CRH^{PVN} neurons that control CORT output. Based on the suggestions of reviewer 2, we have placed less emphasis on the CORT data, using it merely as a guide that directs us to examine CRH^{PVN} cellular activity.

Second, the reviewer is quite correct noting that FS induces a large increase in CRH^{PVN} activity. Importantly, this increase is transient and the activity of CRH^{PVN} neurons, when evaluated in the epochs between foot shocks is not different than activity prior to initiation of the footshocks. We now show these data in Extended Data Fig. 3.

Third, our conclusion is that CRH neurons are activated in both neutral and negative context, but there is a greater increase in activity in the negative context. This increase is a result of recruitment, by foot shock, of neurons that were in a low activity state prior to this stimulus.

2. The authors used CORT as an internal state to study its relationship or coupling with the behavioral state. Also in many figures, they used CORT as an internal index to interpret results and conclude the CORT level outlast the behaviors. But they didn't show any the HPA axis-induced CORT secretion and behavioral changes are independent of each other or that they have a causal relationship somehow, which is a very important premise and question to answer in this study. They should prove whether the uncouple of timescales of the internal and external state is a combinational effect from the circuit and endocrine or just one of them.

As noted above, we have reduced our focus on CORT. We think the main point of this manuscript is that CRH^{PVN} neurons show a form of learning in response to aversive stimuli. This learning, which persists over multiple days must be independent from the context induced behavioral response (freezing) which shows extinction upon repeated exposure. The reviewer is absolutely right that the timescale of the CORT response makes it very hard to set up a causal relationship between the CORT and the behavioral response. Specifically for that reason we applied in vivo recording of PVN CRH neuronal activity and showed the invariability of neuronal activities triggered by multiple re-exposures, that was dramatically different from the changing behavioral responses.

3. The issue of fiber photometry. In most of the recording results, authors showed a period of calcium transient traces before and after stimuli, such as HC vs. NC (Extended Fig. 2d, e, f). Normally, people should show a baseline level of the calcium signals and indicate which time window is used as an offset and what the duration is. Especially sometimes when you do a long-term recording over days, you unplug and plug the ferrules back on the heads, the baseline might change or GCaMP expression intensity might increase gradually, which will have an obvious impact on our results. The authors should show a consecutive trace of each day or indicate how they define the starting times of the home cage and neutral context. I don't see any detailed info in context or even in the method.

We thank the reviewer for bringing up this point. We added illustrative traces for each type of photometry experiment to highlight the intervals of further analysis. Furthermore, we composed an additional extended data figure (Extended Data Fig. 10) to illustrate the recordings. Recordings from subsequent days often lead to a decrease of voltage recorded by the sensor, primarily due to the continuous bleaching of the patch cord. In order to minimize this effect, we performed bleaching correction on the traces from each day. To maximize comparability, we performed z-score calculation on the data that eliminated the recorded voltage differences arising from the setup. Z-score calculation is sensitive to the variability of the baseline but not to the static levels of background autofluorescence. However, to minimize the distortions arising from different baseline activities on different days we used the standard deviation calculated from the baseline period of the first day of the experiment. We clarified and extended the description of the analysis of fiber photometry recordings in the Methods section.

4. As for the mini-scope imaging, authors should indicate the n numbers of mice instead of the cell numbers, or they should increase the quantity for each group to $n \geq 5 - 7$ mice. Also, they should also have some analysis including different individuals to make the results more solid. (e.g. Fig. 2d, Fig.3b)

Currently, there is no standard in the field on minimum number of animals for a given experiment using miniscopes. We added the number of animals used in each miniscope experiments and performed the analysis by collapsing the recordings from individual animals. While for some experiments using miniature microscopes $n=4$ is not necessarily a high number, we would like to emphasize that using fiber

photometry on 10-12 additional animals, we confirmed the main findings from miniscopes. Of course, it was necessary to use miniscopes to obtain cellular resolution, which allowed us to derive a local, novel learning rule based.

5. They showed relatively rough single-cell imaging data to emphasize the CRH neurons' responses to neutral or negative contexts. A principal component analysis shows a shift in the activity state in comparison to HC. I'm curious about if the activity state calculation is based on the overall epoch in HC, neutral context, or footshock chamber. They should make their method more comprehensive and make the legend concise enough. The writing should be improved.

The activity state calculation (via PCA) is based on the aligned data across days, and thus includes the home cage (HC) and neutral context or HC and foot shock chamber, or HC and Hazelnut chamber. To address the referee's criticism, we have made the methods and legends more comprehensive with additional detail, including a detailed section describing how PCA was applied.

6. A follow-up question to comment 5.

What is the correlation between certain behavioral patterns and CRH neuronal activities? Will the behavioral patterns change among different contexts? And what are the calcium signals corresponding to a certain behavioral feature or location in the test arena? In addition to the GCaMP signal changes during the entire stimuli, it'll be more comprehensive and solid if the authors could do a deeper analysis from multiple dimensions.

The reviewer raises an excellent point, however, we feel the deeper analysis of embedded behavioral patterns and their correlation with the activity of individual CRH^{PVN} neurons is beyond the scope of this paper.

7. Based on the scope of this study, the authors demonstrated the behavioral and endocrine responses exhibited distinct timescales for tracking neutral and negative valence. Meanwhile, there's been a surge of studies on CRH neurons' effect on reward processing. In terms of a generic mechanism, is CRH neuron activity involved in associative learning of positive valence? They were unable to provide a generalized working model of CRH neurons, which weakens the significance of CRH in linking external and internal states.

The reviewer puts forward a daunting challenge to provide a generalized working model of CRH neurons. Dozens of labs, over decades, have been studying CRH neurons in the PVN. Based on hundreds of papers, the current working model is: CRH^{PVN} neurons increase activity to aversive stimuli, resulting in an increase in CORT. In the past few years, a handful of papers have now shown a decrease in activity of these cells in response to a positive stimulus. We think our observation that the rapid decrease in CRH^{PVN} activity upon return to the safety of the homecage is consistent with these observations. So, clearly, CRH^{PVN} neurons show bidirectional responses.

Nevertheless, we were intrigued by this provocative comment, so we performed experiments utilizing fiber photometry and miniature microscopy to address whether the memory of appetitive stimulus, associated with a context would impact CRH^{PVN} activity during context re-exposure. And indeed, our new experiments show that re-exposure to a context associated with a positive stimulus elicited a less robust response in CRH^{PVN} neurons (Fig. 5.; Extended Data Fig. 7). Interestingly, the learning rule here is not the reciprocal of that described for aversive stimuli. Instead, we see a simpler 1-factor rule that governs the decrease in the response of CRH^{PVN} neurons upon exposure to the context with positive

association (Extended Data Fig. 8). We have added these exciting new results to the main dataset, extended data figures and the text. This expands our manuscript in an exciting new direction.

8. In the conclusion part, by proposing a learning rule which might add a new dimension to classical learning theory, I don't see more other novelties or contributions of this paper to this field beyond the above statement. And the amount of evidence and results seems weak to me. Overall, it looks like an unestablished work needs to be improved a lot. I don't think it's eligible for publication at this stage.

We are pleased the reviewer appreciates the novelty of the learning rule.

Minor comments:

1. Method needs to be supplemented with more details.
2. Authors might need to include all the individual points in all graphs in addition to mean \pm s.e.m. information. (e.g. Fig. 1j, 5d)

The mentioned graphs show the means of individual data points that are already shown in other graphs.

3. The same types of graphs have different displays in all figures, try to make the style as one to make it more concise (Fig. 1a, 2c, 4c).

The mentioned figures show different types of data, behavior, individual PVN CRH neuronal activity and simulated neuronal activity, respectively. For that reason, we believe that making these graphs uniform might confuse the reader. As we reconstructed the figures, we paid extra attention to show behavioral, miniscope and photometry data differently but in a consistent manner.

Reviewer #2 (Remarks to the Author):

The paper is interesting and the experiments are conducted with a high technical standard. However, I disagree with the interpretation of the data and I believe additional experiments would be needed to substantiate the conclusions.

a. A main criticism is the focus of the authors on negative stimuli. CRH-PVN neurons and the HPA axis are not only activated by stimuli with negative valence. In contrast, there is ample evidence that also stimuli with positive valence that lead to positive emotions like joy or excitement significantly activate the HPA axis and CORT secretion. Due to this indiscriminate nature of HPA axis activity, and therefore also CRH-PVN neuronal activity, I would argue that both are unreliable indicators of emotion, independent of the detail and time-scale of the activity assessment.

The reviewer raises a very important issue here and these comments forced us to look very seriously at using emotions as a focal point for the manuscript, *and* then, CORT as a proxy for emotions. Consequently, we have removed most of the CORT data from the body of the manuscript. We have a differing opinion, however, on the statement that CRH-PVN neuronal activity is indiscriminate. There is now compelling evidence that CRH-PVN activity increases in response to aversive stimuli (Kim et al, 2018, Daviu et al, 2020) and decreases in response to appetitive stimuli (Kim et al, 2018). This bidirectionality allowed us to ask direct questions about how individual cells respond to, and store information about stimuli of different valence.

A main assumption of the authors is that CRH-PVN activity is reflective of the valence of internal states, but they only show this with different levels of negative valence. I believe it would be important to test how a positive stimulus (e.g., a reward) would affect behavior in a neutral context and how this would relate to CORT secretion and CRH-PVN activity.

As described in response to R1 and the first comment to R2, we have now examined responses to an appetitive stimulus (Nutella) in a neutral context. We report a decrease in the activity of CRH-PVN neurons, confirming the findings of Kim et al. We then went on to ask whether this would be sufficient to change future contextual responses. Remarkably, exposure to an appetitive stimulus resulted in a decreased response of CRH-PVN neurons to this context. We also show that this decrease does not rely on a 2-factor learning rule as we describe for aversive learning, but rather a 1-factor rule in which cells all decrease their activity in the context.

b. The authors interpret the absence of CORT habituation to the neutral environment as dissociation from the behavioral habituation. However, the CORT response is likely triggered just as much by the disturbance of the animals from their home cage environment and handling by the investigator. Thus, while the memory to the already explored environment will result in behavioral adaptation, the remaining threat of being disturbed and handled by the experimenter will always result in a similar CORT response, just as the regular cage change would. I am not sure how this can be tested experimentally, but it calls into question the main hypothesis and interpretation of the authors. Would the same CRH-PVN cells that respond to a neutral or aversive context not also respond in parallel to the circadian activity of the HPA axis in a home cage setting?

The reviewer makes an excellent point and also understands the difficulty of resolving this issue. In the absence of having a real-time readout of CORT, we decided to focus on real-time activity of CRH-PVN neurons. We conducted additional experiments in four animals where we used miniscopes to evaluate CRH-PVN activity during pickup and return to the homecage. We show in Extended Data Fig. 2, that this manipulation results in an increase in CRH-PVN activity that is transient, recovering to baseline levels within seconds of releasing the animal back into the homecage. This is at odds with the observation that placement in a neutral environment results in an elevation in CRH-PVN activity that persists for the duration of the exposure and only returns to baseline levels upon return to homecage.

c. Can the authors track if weak and strong CRH-PVN cells have different inputs and are preferentially innervated by different cell populations / brain regions? Mapping this potentially differential input network could result in additional insights in the contribution of these two neuronal sub-populations in encoding stressful stimuli.

This is an important question, however the tools necessary for the combination of tract tracing and post hoc identification of these neurons following the application of miniature microscopy are not available. Furthermore, studies involving the inputs of the phenotypes of CRH neurons (Romanov et al., 2017, Nat Neurosci; Kondoh et al., 2016, Nature) show no distinct CRH subpopulations, rather a large diversity of cells expressing CRH. In line with this, our analysis including the whole recorded population showed a spectrum of responses (Extended Data Fig. 5).

REVIEWER COMMENTS

Reviewer #1 (Remarks to the Author):

In the revised manuscript, the authors have performed additional analyses to support the hypothesis proposed in the first version of their work. However, I still have reservations about the conceptual aspects that I brought up in my initial review.

1. In my original assessment, I questioned whether the lack of frequent sampling could potentially mask the kinetics of CORT secretion. The authors responded by shifting their focus from CORT to the activity of CRH neurons. This adjustment is puzzling, especially given that the activity of CRH neurons did not show a significantly different response pattern like that of CORT. This is particularly concerning because their original manuscript emphasized distinct timescales in tracking neutral and negative valence for behavioral and endocrine responses. This alteration diminishes the overall impact and interest of the manuscript.

2. Besides conceptual issues that are still present, despite the revision, I am concerned about methodological and technological aspects. Previously, I pointed out the difficulty of comparing fiber photometry signals across multi-day recordings due to issues like unplugging and plugging ferrules and varying GCaMP expression and signal efficiency. Their reliance on Z-score calculations does not adequately address these problems. To strengthen their methodology, the authors should consider recording the signal continuously over multiple days without disconnecting and reconnecting the ferrules. They should also aim to verify and calibrate GCaMP signals suitable for such long-term recordings.

3. A deeper analysis of embedded behavioral patterns and their correlation with the activity of individual CRH neurons could strengthen the manuscript. Without such an in-depth analysis, the manuscript risks falling short of providing a complete picture of the relationship between behavioral patterns and CRH neuronal activity.

Reviewer #2 (Remarks to the Author):

The authors thoughtfully addressed all of my previous points and added convincing additional data. Following this revision have have no further criticism and support publication of the manuscript in its current form.

1. In my original assessment, I questioned whether the lack of frequent sampling could potentially mask the kinetics of CORT secretion. The authors responded by shifting their focus from CORT to the activity of CRH neurons. This adjustment is puzzling, especially given that the activity of CRH neurons did not show a significantly different response pattern like that of CORT. This is particularly concerning because their original manuscript emphasized distinct timescales in tracking neutral and negative valence for behavioral and endocrine responses. This alteration diminishes the overall impact and interest of the manuscript.

The lack of frequent sampling could potentially mask CORT release kinetics. It is also true that CORT does not provide a direct readout of CRH activity. Nevertheless, even if we did sample CORT more frequently, it would still not approach the temporal resolution achieved by in vivo Ca imaging. Furthermore, as mentioned in the paper, the level of CORT release is also subject to peripheral adjustments, making CORT measurements unreliable. Finally, we note, that one of our findings is that this system is exquisitely sensitive to small perturbations (the experimenter entering the room, picking up the mouse even after dozens of handlings). As a consequence, multiple samplings will have a significant impact on the recorded activity and CORT release in mice (Kim et al., *Steroids*, 2018).

2. Besides conceptual issues that are still present, despite the revision, I am concerned about methodological and technological aspects. Previously, I pointed out the difficulty of comparing fiber photometry signals across multi-day recordings due to issues like unplugging and plugging ferrules and varying GCaMP expression and signal efficiency. Their reliance on Z-score calculations does not adequately address these problems. To strengthen their methodology, the authors should consider recording the signal continuously over multiple days without disconnecting and reconnecting the ferrules. They should also aim to verify and calibrate GCaMP signals suitable for such long-term recordings.

We appreciate the reviewer's point about comparing fiber photometry recordings from multiple days. This can, indeed be challenging, although it has been performed by multiple labs, including ours (Li et al., *Neuron*, 2023; Suthard et al., *J Neurosci*, 2023; Daviu et al., *Nat Neurosci*, 2020). We added further explanation on page 12 of the manuscript: "On each day a 3-min baseline period in the home cage prior placing the animal to a different context was used to calculate relative change triggered by context exposure.", to emphasize that we evaluated a change relative to home cage baseline on each day. Analyses can either use relative values (such as dF/F or z-score calculated for each day and animal) or absolute values (such as dF or z-score calculated for the animal). The former approach is insensitive to issues such as increase in GCaMP expression or altered throughput in the light pathway but is sensitive to differences imposed by different levels of fiber autofluorescence. The latter approach has the opposite strengths and limitations. To address the reviewer's concerns, and our own curiosity, we repeated the analysis using z-score based on standard deviation from each day. We reached the same conclusion and added the following text to the methods on page 12: To exclude the possibility that slow changes in GCaMP expression or basal CRH^{PVN} activity that affect both baseline and context exposure recording caused the observed changes during repeated exposures we performed the fiber photometry analyses using methods that are insensitive to the baseline and use relative differences exclusively. Photometry dF/F was calculated by using least-squares second order polynomial fit we fitted the 405 nm channel to the 465 nm channel. Then performed the following calculation: $dF/F = (F_{465} - F_{\text{fitted } 405}) / F_{\text{fitted } 405}$. Additionally, we performed the formerly described z-score analysis using the standard deviation calculated from the home cage baseline activity. All analyses led to the same conclusions (Extended Data Fig. 10b-i). We added the results to Extended Fig. 10. Finally, although the reviewer's suggestion of

continuous sampling is theoretically possible, the continuous delivery of light with result in significant photobleaching and pose even more challenges for GCaMP signal calibration.

3. A deeper analysis of embedded behavioral patterns and their correlation with the activity of individual CRH neurons could strengthen the manuscript. Without such an in-depth analysis, the manuscript risks falling short of providing a complete picture of the relationship between behavioral patterns and CRH neuronal activity.

We agree that this is a very interesting question, but is beyond the scope of this manuscript which focused on the existence of a hypothalamic memory center encoding contextual valence. Neither the lack or the presence of the specific CRH subpopulation firing pattern correlating with a behavioral activity would change the memory encoded in the mean of the activity. The highly overlapping state spaces occupied by CRH activity during repeated recalls suggest that either 1) the behaviors correlating with CRH activity are very rare during the fear extinction paradigm, 2) the correlating CRH activity is very subtle or 3) there is no correlation whatsoever. Providing a complete picture about the link between behavior and embedded patterns of CRH network activity is a laudable goal and perhaps best pursued in multiple future studies from our lab and/or others.

REVIEWERS' COMMENTS

Reviewer #1 (Remarks to the Author):

While my concerns persist, I appreciate the author's responses addressing my previous queries. As a result, I am inclined to support the publication of this work in Nature Communications.